# A Generative Nonparametric Bayesian Model for Whole Genomes

**Alan N. Amin**[*1,2], **Eli N. Weinstein**[*2,3] **and Debora S. Marks**[2,4]

[1] Program in Systems, Synthetic and Quantitative Biology
[2] Department of Systems Biology
Harvard Medical School
[3] Program in Biophysics, Harvard University
[4] Broad Institute of Harvard and MIT
alanamin@fas.harvard.edu, eweinstein@g.harvard.edu, debbie@hms.harvard.edu

## Abstract

Generative probabilistic modeling of biological sequences has widespread existing and potential use across biology and biomedicine, particularly given advances in high-throughput sequencing, synthesis and editing. However, we still lack methods with nucleotide resolution that are tractable at the scale of whole genomes and that can achieve high predictive accuracy in theory and practice. In this article we propose a new generative sequence model, the Bayesian embedded autoregressive (BEAR) model, which uses a parametric autoregressive model to specify a conjugate prior over a nonparametric Bayesian Markov model. We explore, theoretically and empirically, applications of BEAR models to a variety of statistical problems including density estimation, robust parameter estimation, goodness-of-fit tests, and two-sample tests. We prove rigorous asymptotic consistency results including nonparametric posterior concentration rates. We scale inference in BEAR models to datasets containing tens of billions of nucleotides. On genomic, transcriptomic, and metagenomic sequence data we show that BEAR models provide large increases in predictive performance as compared to parametric autoregressive models, among other results. BEAR models offer a flexible and scalable framework, with theoretical guarantees, for building and critiquing generative models at the whole genome scale.

## 1 Introduction

Measuring and making DNA is central to modern biology and biomedicine. Generative probabilistic modeling offers a framework for learning from sequencing data and forming experimentally testable predictions of unobserved or future sequences that can be synthesized in the laboratory [19, 31, 63]. Existing approaches to genome modeling typically preprocess the data to build a matrix of genetic variants such as single nucleotide polymorphisms [25, 57]. However, most modes of sequence variation are more complex. Structural variation occurs widely within individuals (e.g. in cancer), between individuals (e.g. in domesticated plant populations) and between species (e.g. in the human microbiome), and methods for detecting and classifying structural variants are heuristic and designed only for predefined types of sequence variation such as repeats [12, 44, 50, 69, 78]. Ideally, we would be able to directly model genome sequencing data and/or assembled genome sequences. However, building generative models that work with raw nucleotides, not matrices of alleles, raises the

---

[*]These authors contributed equally.

35th Conference on Neural Information Processing Systems (NeurIPS 2021).

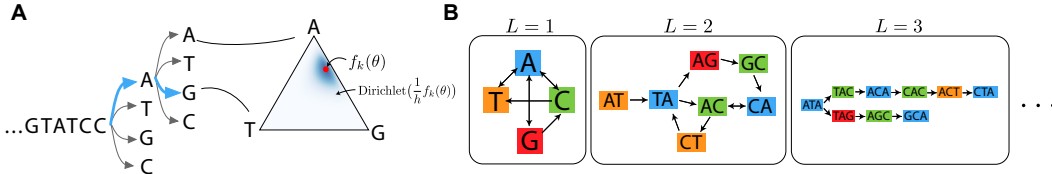

Figure 1: **Overview of the BEAR model.** (A) BEAR models employ a Dirichlet prior on Markov transition probabilities that is centered at the prediction of an AR model. (B) De Bruijn graphs showing BEAR transitions with non-zero probability under an example data-generating distribution. As the lag $L$ increases, the model has higher resolution.

extreme statistical challenges of having enough *flexiblility* to account for genomic complexity, *interpretability* to reach scientific conclusions, and *scalability* to train on billions of nucleotides. Given the relevance of genetic analysis to human health, models should also possess strong *theoretical guarantees.*

Autoregressive (AR) models are a natural starting point for generative genome modeling, since they (1) have been successfully applied to biological sequences, as well as many other types of non-biological sequential data, (2) can be designed to have interpretable parameters, and (3) can be scaled to big datasets with very long sequences [66, 73]. However, since AR models are parametric models, they will in general suffer from misspecification; as we show empirically in Section 6, for genomic datasets misspecification can be a serious practical limitation not only for simple AR models but even for deep neural networks.

As an alternative strategy for building generative probabilistic models at the genome scale, we propose in Section 2 the nonparametric "Bayesian embedded autoregressive" (BEAR) model. BEAR models are Bayesian Markov models, with a prior on the lag and conjugate Dirichlet priors on the transition probabilities. The hyperparameters of the Dirichlet prior are controlled by an "embedded" AR model with parameters $\theta$ and an overall concentration hyperparameter $h$, both of which can be optimized via empirical Bayes. In Section 3 we show that BEAR models can capture arbitrary data-generating distributions, and establish asymptotic consistency guarantees and convergence rates for nonparametric density estimation. In Section 4, we show that the optimal $h$ provides a diagnostic for whether or not the embedded AR model is misspecified and if so by how much, alerting the practitioner when the parameter estimates $\theta$ are untrustworthy. Besides estimation problems, BEAR models can also be used to construct goodness-of-fit tests and two-sample tests, thanks to their analytic marginal likelihoods, and we prove consistency results for these tests in Section 5. Finally we apply BEAR models at large scale, to genomic datasets with tens of billions of nucleotides, including whole genome, whole transcriptome, and metagenomic sequencing data; we find that BEAR models can have greatly improved performance over AR models (Section 6).

Crucial to our theoretical and empirical analysis is the statistical setting: we assume that the data $X_1, \ldots, X_N$ consists of finite but possibly variable length strings (with small alphabets) drawn i.i.d. from some underlying distribution $p^*$, and study the behavior of estimators and tests as $N \to \infty$. This setup differs from common theoretical analyses of sequence models outside of biology, which typically consider the limit as the length of an individual sequence goes to infinity [26]. In biology, however, we observe finite sequences recorded from many individual species, organisms, cells, molecules, etc. and want to generalize to unseen sequences, making $N \to \infty$ the appropriate large data limit.

## 2   Bayesian embedded autoregressive models

We first briefly review autoregressive (AR) models as applied to sequences of discrete characters. Let $f(\theta)$ denote an autoregressive function with parameter $\theta$ and let $L$ denote the lag of the autoregressive model; then the AR model generates data as

$$X_i | X_{i-L:i-1} \sim \text{Categorical}(f_{X_{i-L:i-1}}(\theta)), \tag{1}$$

where $i$ indexes position in the sequence $X$ and $X_{i-L:i-1}$ consists of the previous $L$ letters in the sequence. Since sequence length as well as nucleotide or amino acid content is relevant to biological applications, we use a start symbol $\emptyset$ at the beginning and a stop symbol \$ at the end of each sequence; letters $X_i$ are sampled sequentially starting from the start symbol and continuing until a stop symbol is drawn.

We propose the Bayesian embedded autoregressive (BEAR) model, a Bayesian Markov model that embeds an AR model into its prior. The BEAR model takes the form,

$$L \sim \pi(l), \quad v_k \sim \text{Dirichlet}\left(\frac{1}{h} f_k(\theta)\right) \text{ for all } k,$$

$$X_i|X_{i-L:i-1} \sim \text{Categorical}(v_{X_{i-L:i-1}}),$$

(2)

where $\pi(l)$ is a prior on the lag with support up to infinity, $h > 0$ is a concentration hyperparameter, and $k$ is a length $L$ kmer. The BEAR model has three key properties (Fig. 1). First, the unrestricted transition parameter $v$ and lag $L$ allow the model to capture exact conditional distributions of $p^*$ to arbitrarily high order: $p^*(X_i|X_{i-1})$ at $L = 1$, then $p^*(X_i|X_{i-2}, X_{i-1})$ at $L = 2$, etc.. This property allows the BEAR model to be used for nonparametric density estimation (Section 3). Second, in the limit where $h \to 0$, the BEAR model reduces to the embedded AR model (Eqn. 1). The optimal $h$ provides a measurement of the amount of misspecification in the AR model (Section 4). Third, the choice of the conjugate Dirichlet prior allows the conditional marginals $p((X_n)_{n=1}^N|L, h, \theta)$ to be computed analytically, and (since $L$ is one-dimensional) the total marginal likelihood $p((X_n)_{n=1}^N|h, \theta)$ to be estimated tractably. This allows BEAR models to be used for hypothesis testing (Section 5).

There are a variety of ways of performing inference in BEAR models, but for most applications we will focus on empirical Bayes methods that optimize point estimates of $L$, $h$ and $\theta$. Let $\#(k, b)$ denote the number of times the length $L$ kmer $k$ is seen followed by the letter or stop symbol $b$ in the dataset $(X_n)_{n=1}^N$. Using a high-performance kmer counter optimized for nucleotide data, KMC, we can compute the count matrix $\#(\cdot, \cdot)$ for all observed kmers $k$ in terabyte-scale datasets, even when the matrix does not fit in main memory (Section J.2) [39]. To optimize $h$ and $\theta$, we take advantage of the fact that the log conditional marginal likelihood can be written as a sum over observed kmers,

$$\log p((X_n)_{n=1}^N|L, h, \theta) = \sum_{k:\#k>0} \log\left[\frac{\Gamma(\sum_b \frac{1}{h} f_{kb}(\theta))}{\prod_b \Gamma(\frac{1}{h} f_{kb}(\theta))} \frac{\prod_b \Gamma(\frac{1}{h} f_{kb}(\theta) + \#(k, b))}{\Gamma(\sum_b \frac{1}{h} f_{kb}(\theta) + \#(k, b))}\right].$$

(3)

This decomposition lets us construct unbiased stochastic estimates of the gradient with respect to $h$ and $\theta$ by subsampling rows of the count matrix (Section J.1). Empirical Bayes in the BEAR model therefore costs little extra time as compared to standard stochastic gradient-based optimization of the original AR model. Code is available at https://github.com/debbiemarkslab/BEAR.

**Toy example** We next briefly illustrate the properties and advantages of the BEAR model in simulation. We generated samples from an AR model in which $f_k(\theta)$ depends on $k$ linearly as a function of both individual positions and pairwise interactions between positions, with the strength of the pairwise interaction weighted by a parameter $\beta^*$ (Section I.1). We first fit (using maximum likelihood) a linear AR model that lacks pairwise terms and is thus misspecified when $\beta^* > 0$. Since the AR model is misspecified, it does not asymptotically approach the true data-generating distribution $p^*$ (Fig. 2A, gray). We next computed the posterior of a vanilla BEAR model without the embedded AR in its prior, instead using the Jeffreys prior $v_k \sim_{iid} \text{Dirichlet}(1/2, \ldots, 1/2)$. The vanilla BEAR model asymptotically approaches the true data generating distribution, since it is a nonparametric model; however, it underperforms the AR model in the low data regime (Fig. 2A, black). Finally, we fit a BEAR model with the misspecified linear AR model embedded, using our empirical Bayes procedure. The BEAR model performs just as well as its embedded AR model in the low data regime, just as well as the vanilla model in the high data regime, and better than both at intermediate values (Fig. 2A, blue and yellow).

When the AR model is well-specified, the empirical Bayes estimates of the parameters $\theta$ under the BEAR model match the maximum likelihood estimates of $\theta$ under the AR model

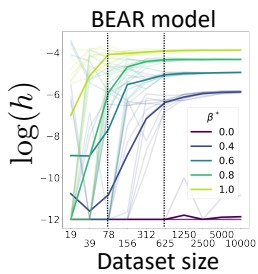

**A. Density estimation**

$\beta^* = 0.0$
Well-specified

$\beta^* = 0.6$
Low misspecification

$\beta^* = 1.0$
High misspecification

**B. Diagnosing misspecification**

BEAR model

Figure 2: **BEAR models detect and avoid misspecification without sacrificing small dataset performance.** (A) Estimated KL divergence between simulated data-generating distribution $p^*$ and model posterior predictive distribution, as a function of dataset size $N$. Five independent simulations were run; thin lines correspond to individual simulations, thick lines to the average across simulations. (B) The $h$ misspecification diagnostic as a function of dataset size, for varying $\beta^*$. Dataset sizes at which $h$ is close to convergence for $\beta^* = 0.6$ (right) and $\beta^* = 1.0$ (left) are marked with vertical lines.

nearly exactly (Fig. S7). When the AR model is misspecified, however, the BEAR model provides a warning: the empirical Bayes estimate of $h$ converges to a non-zero value, rather than zero (Fig. 2B). This warning emerges early: $h$ converges well before the vanilla model starts outperforming the misspecified AR model.

**Related Work** The key idea behind BEAR models is to nonparametrically perturb a parametric model [48], following a similar strategy to the Polya tree method proposed by Berger and Guglielmi [6]. As in Berger and Guglielmi [6], we use Dirichlet priors centered at the parametric model's predictions, and construct tractable goodness-of-fit tests by exploiting Dirichlet-categorical conjugacy. BEAR models extend these ideas from one-dimensional continuous data to finite-length sequences of discrete characters.

Markov and AR models have a long history and wide range of applications in biological sequence analysis [21, 51, 58]. Compression methods, in particular, often rely on accurate density estimation and use Markov or AR models to achieve it [17, 55, 56, 67]. We establish theoretical guarantees for density estimation with fully Bayesian Markov models (Section 3). AR models used for compression, like other AR models, can be embedded into BEAR models for improved statistical performance and to measure misspecification.

BEAR models are closely linked to non-generative genome analysis methods. Assembly algorithms and variant callers often analyze paths in the de Bruijn graph of a sequence dataset; in the limit $h \to \infty$, samples from the posterior predictive distribution of the BEAR model, conditional on $L$, correspond to paths through the $L$-mer de Bruijn graph of the data [11, 33]. Comparisons between genomes and other sequences are often made on the basis of kmer counts; our two-sample test provides a generative perspective on this idea [3, 16, 78].

BEAR models are also connected to ideas in natural language processing, where kmers are referred to as ngrams. Under the vanilla BEAR model, the mean of the posterior predictive distribution conditional on $L$ corresponds to an ngram additive smoothing model [9]. Comparisons between datasets using their ngram counts are also common in model evaluation metrics such as the BLEU score [53].

## 3    Density estimation

The density estimation problem is that of estimating $p^*$ given data $(X_n)_{n=1}^N$ drawn i.i.d. from $p^*$. Density estimation is particularly crucial for biological sequence analysis due to its connections to fitness estimation [31, 65]. State-of-the-art mutation effect prediction methods and clinical variant interpretation methods rely on density estimates of evolutionary sequence data [20, 60]. Density estimation with generative models is particularly useful for protein design, as samples from accurate density estimates are likely to be functional and

can be synthesized in the laboratory [63, 66]. Despite all these applications, existing density estimation methods for biological sequences lack theoretical guarantees on their accuracy and are often limited in their scale, being restricted to relatively short sequences [79]. Here, we show that the posterior distribution of the BEAR model is consistent and will concentrate on $p^*$ as $N \to \infty$, regardless of what $p^*$ actually is, so long as $p^*$ generates finite length sequences almost surely (a.s.).

We first study the expressiveness of BEAR models. Let $\mathcal{M}_L$ be the set of Markov models $p_v$ with transition probabilities $v$ and lag $L$ that generate finite length strings a.s.. Note that $\mathcal{M}_1 \subset \mathcal{M}_2 \subset \ldots$. Define the union $\mathcal{M} = \cup_{L=1}^\infty \mathcal{M}_L$. We can compare $\mathcal{M}$ to the set of distributions over finite strings $S$, of which $p^*$ is a member. In Section D we prove that,

**Summary of Propositions 1-4** *Not all possible distributions over $S$ are in $\mathcal{M}$. However, $\mathcal{M}$ is dense on the space of probability distributions over $S$ with the total variation metric.* The implication of this result is that although BEAR models cannot exactly match arbitrary data-generating distributions, they can approximate $p^*$ arbitrarily well as $L$ increases. This makes asymptotic consistency possible.

We now show that the posterior of the BEAR will in fact asymptotically concentrate on the true $p^*$, i.e. it is consistent. For tractability, we assume in this section that the prior is fixed (we do not use empirical Bayes). The result relies on the tools for understanding convergence rates of posteriors developed in Ghosal et al. [23]. The most important assumption is that $p^*$ is subexponential, meaning that for some $t > 0$, $E_{p^*} \exp(t|X|) < \infty$ where $|X|$ is the sequence length. Let $\Pi(\cdot|(X_n)_{n=1}^N)$ denote the posterior over sequence distributions. Let $B(p^*, \delta)$ denote a ball of radius $\delta$ centered at $p^*$, using the Hellinger distance.

**Summary of Theorem 35** *Given $M > 0$ large enough and $\epsilon \in (0, 1)$ small enough, we have $\Pi(B(p^*, MN^{-\frac{1}{2}\epsilon})|(X_n)_{n=1}^N) \to 1$ in probability.*

A proof is in Section H and simulations in Section I.2. This result states that the posterior distribution of the model converges to a delta function at the true distribution $p^*$ regardless of what $p^*$ is. It also provides a rate of convergence: in a parametric model, the uncertainty would shrink as $N^{-\frac{1}{2}}$, but here the rate is slower, $N^{-\frac{1}{2}\epsilon}$, a price paid for the nonparametric model's expressivity [23, 29, 35]. The proof includes a variety of new theoretical constructions and algorithms that are used to approximate subexponential sequence distributions.

## 4  Robust parameter estimation

To derive a biological understanding of mutational processes, evolutionary history, functional constraints, etc. from sequence data, researchers must estimate model parameters (not just density). However, parameter estimates cannot in general be trusted when models are misspecified [34]. To reach robust scientific conclusions, therefore, parameter estimates should ideally come with a warning about whether or not the model is misspecified and some measurement of the degree of misspecification. Here, we study in BEAR models the asymptotic behavior of empirical Bayes estimates of the AR parameter $\theta$, as well as the hyperparameter $h$, showing that $h$ diagnoses misspecification in the embedded AR model.

Our analysis builds off the study of empirical Bayes consistency in Petrone et al. [54], which showed that empirical Bayes will, in general, maximize the prior probability of the true data-generating parameter value. Extending this theory to BEAR models is nontrivial, since in BEAR models the standard Laplace approximation to the marginal likelihood can fail. For theoretical tractability, as in many analyses of similar models, we fix $L$ at some arbitrary and large value [30]. Define $p^{*(L)} = \mathrm{argmin}_{p_v \in \mathcal{M}_L} \mathrm{KL}(p^*\|p_v)$ as the closest model in $\mathcal{M}_L$ to $p^*$, and define $v^*$ such that $p_{v^*} = p^{*(L)}$ (note $p^{*(L)} \to p^*$ as $L \to \infty$). We say that the AR model is misspecified "at resolution $L$" if $f$ cannot approximate $p^{*(L)}$, i.e. if there does not exist some sequence of parameter values $\tilde{\theta}_N$ such that $p_{f(\tilde{\theta}_N)} \to p^{*(L)}$ as $N \to \infty$; otherwise, the AR model is well-specified at resolution $L$. Now we can study empirical Bayes estimates of $h$ and $\theta$, denoted $h_N$ and $\theta_N$.

**Summary of Propositions 15-20** *Let $(h_N)_{N=1}^\infty$ and $(\theta_N)_{N=1}^\infty$ be sequences maximizing the BEAR marginal likelihood $p((X_n)_{n=1}^N|L, h, \theta)$ for each $N$. If the model is well-specified at resolution $L$, then $h_N N^{1/4-\epsilon} \to 0$ for every $\epsilon > 0$ and $p_{f(\theta_N)} \to p^{*(L)}$ in distribution, with both sequences converging in probability. On the other hand, if the model is misspecified at*

*resolution L, then $h_N$ is eventually bounded below by some positive (non-zero) number a.s..* Proofs are in Section F and simulations in Section I.1. The implication of this result is that when the AR model is well-specified, $h_N$ converges to zero (at a rate that is a power of the dataset size) and $\theta_N$ converges to the parameter value $\theta^*$ at which the AR model matches the data (Corollary 16). On the other hand, when the AR model is misspecified, $h_N$ does not converge to zero; heuristically, we find instead that $h_N$ is approximately proportional to a divergence between $p^{*(L)}$ and the AR model,

$$h_N \propto \sum_{k \in \mathrm{acc}_L(p^*)} \left( \mathrm{KL}(f_k(\theta_N) \| v_k^*) + \log(N) \sum_{b \notin \mathrm{supp}_L(p^*)|_k} f_{k,b}(\theta_N) \right), \tag{4}$$

where $\mathrm{acc}_L(p^*) = \{k \mid p^*(\#k > 0) > 0\}$ is the set of kmers with non-zero probability and $\mathrm{supp}_L(p^*)|_k = \{b \mid p^*(\#(k,b) > 0) > 0\}$ is the set of transitions from $k$ with non-zero probability. In summary: when fitting a BEAR model by empirical Bayes, you get, along with a parameter estimate $\theta_N$, a value $h_N$ which tells you the amount (from zero to infinity) of misspecification in the AR model. If $h_N$ is close to zero, you can trust the estimate $\theta_N$.

## 5 Hypothesis testing

**Goodness-of-fit test** A major outstanding challenge in biological sequence analysis is to build models based on natural sequence data that are accurate enough to generate novel functional sequences [45]. A crucial component of the problem is model evaluation: while relative model performance may be compared on the basis of likelihood, absolute performance – whether or not the model in fact provides an accurate description of the data – is usually addressed solely on the basis of limited numbers of summary statistics, such as average amino acid hydrophobicity or sequence length [63, 66]. Given a dataset $(X_n)_{n=1}^N \sim p^*$ i.i.d., a goodness-of-fit test asks whether or not the data distribution $p^*$ matches a model distribution $\tilde{p}$. It takes into account all possible distributions $p^*$ including those that differ from $\tilde{p}$ in a manner that cannot be captured by finitely many summary statistics. We propose a goodness-of-fit test that compares the null hypothesis $\mathcal{H}_0 : p^* = \tilde{p}$ to the alternative $\mathcal{H}_1 : p^* \neq \tilde{p}$ using the Bayes factor $\mathrm{BF} = p((X_n)_{n=1}^N | h, \theta) / \tilde{p}(X_{1:n})$, where $p((X_n)_{n=1}^N | h, \theta) = \sum_L p((X_n)_{n=1}^N | L, h, \theta) \pi(L)$ is the marginal likelihood under the BEAR model. Note that practically, the sum over $L$ is straightforward to approximate by truncation, and that the test can be computed in time linear in the amount of data.

We now prove the consistency of the test. As in comparable theoretical analyses of tests based on Polya trees, for theoretical tractability we truncate the prior, setting $\pi(L) = 0$ for $L$ larger than some arbitrary $\tilde{L}$ but $\pi(L) > 0$ for $L \leq \tilde{L}$ [30]. We treat $\theta$ and $h > 0$ as fixed. **Summary of Proposition 21** *If $\tilde{p}$ is at least as close to $p^*$ as $p^{*(L)}$ is, as measured by* $\mathrm{KL}(p^* \| \cdot)$, *then* $\mathrm{BF} \to 0$ *in probability as $N \to \infty$. On the other hand, if $p^{*(L)}$ is closer than $\tilde{p}$, then* $\mathrm{BF} \to \infty$ *in probability.* A proof is in Section G.1 and simulations in Section I.3.

An important practical limitation on nonparametric hypothesis testing is low power: since so many alternative distributions must be considered, the null hypothesis can rarely be rejected. However, Proposition 21 holds for the Bayes factor $\mathrm{BF}(L, h, \theta) = p((X_n)_{n=1}^N | L, h, \theta) / \tilde{p}((X_n)_{n=1}^N)$ with any choice of $L$, $h > 0$, and $\theta$. Thus in practice to increase power we can maximize the value of $\mathrm{BF}(L, h, \theta)$ as a function of $L$, $h$, and/or $\theta$ (note that this approach is heuristic, since we have not proven the consistency of the maximized Bayes factor). Berger and Guglielmi [6] provide extensive methodological guidance on using analogous tests constructed with Polya trees. Based on their recommendations, we suggest first choosing $\theta$ such that $p_{f(\theta)}$ is as close as possible to $\tilde{p}$, then plotting the Bayes factor as a function of $h$ and/or $L$ to identify the maximum value and confirm that any conclusion is robust to changes in $h$ and/or $L$.

Another challenge in nonparametric hypothesis testing is that it can be difficult to understand how exactly a test reached its conclusion. To identify which sequences provided the most evidence for or against the null hypothesis, we suggest examining the BEAR Bayes factor for each individual sequence conditional on the rest of the dataset, in analogy to the witness function used in kernel-based tests [43, 70].

Table 1: **Heldout perplexity.** *Whole genome sequencing data*: YSD1: A Salmonella phage. *A. th.*: *Arabidopsis thaliana*, a plant (datasets represent different individuals). *Single cell RNA sequencing data*: PBMC: peripheral blood mononuclear cells, taken from a healthy donor. HL: Hodgkin's lymphoma tumor cells. GBM: glioblastoma tumor cells. *Metagenomic sequencing data*: HC: non-CD and non-UC controls. CD: Crohn's disease. UC: ulcerative colitis. *Full assembled genomes*: Bact.: Bacteria. *Models* Van.: Vanilla (Jeffreys prior). Lin.: Linear. CNN: convolutional neural network. Ref.: reference genome/transcriptome model.

| Dataset | AR Lin. | AR CNN | AR Ref. | BEAR Van. | BEAR Lin. | BEAR CNN | BEAR Ref. |
|---|---|---|---|---|---|---|---|
| YSD1 | 3.953 | 3.873 | 1.266 | 1.165 | **1.144** | **1.144** | 1.145 |
| *A. th.* 1 | 3.956 | 3.947 | 2.686 | 1.567 | 1.432 | 1.432 | **1.411** |
| *A. th.* 2 | 3.953 | 3.949 | 1.982 | 1.650 | 1.463 | 1.462 | **1.441** |
| *A. th.* 3 | 3.998 | 3.952 | 2.340 | 1.834 | 1.728 | **1.727** | 1.733 |
| PBMC | 3.991 | 3.974 | 2.097 | 1.402 | **1.372** | **1.372** | 1.374 |
| HL | 3.959 | 3.930 | 2.141 | 1.409 | **1.378** | **1.378** | 1.379 |
| GBM | 4.137 | 4.137 | 2.366 | 1.442 | **1.406** | **1.406** | **1.406** |
| HC | 3.966 | 3.946 | - | 1.652 | 1.465 | **1.464** | - |
| CD | 3.992 | 3.985 | - | 1.760 | **1.524** | **1.524** | - |
| UC | 3.989 | 3.986 | - | 1.644 | **1.481** | **1.481** | - |
| Bact. | 3.831 | 3.794 | - | **3.774** | **3.774** | **3.774** | - |

**Two-sample test** A two-sample test asks whether or not two datasets $(X_n)_{n=1}^N$ and $(X'_n)_{n=1}^{N'}$ are drawn from the same distribution. Efforts to compare different sequence datasets are widespread in biology: for instance, researchers often wish to determine whether two microbiome samples, taken under different conditions or at different timepoints, are the same up to sampling noise [44]. Two-sample tests can also be used to evaluate generative sequence models that lack tractable likelihoods (for which the goodness-of-fit test proposed above does not apply) such as energy-based models or implicit models like GANs and biophysical simulators [27, 42, 49]. Assume $(X_n)_{n=1}^N \sim p_1$ and $(X'_n)_{n=1}^{N'} \sim p_2$ i.i.d.. Our BEAR test compares the null hypothesis $\mathcal{H}_0 : p_1 = p_2$ to the alternative $\mathcal{H}_1 : p_1 \neq p_2$ using the Bayes factor $\text{BF} = p((X_n)_{n=1}^N|h, \theta)p((X'_n)_{n=1}^{N'}|h, \theta)/p((X_n)_{n=1}^N, (X'_n)_{n=1}^{N'}|h, \theta)$. As in the goodness-of-fit case, the test can be computed approximately in time linear in the amount of data, and the same advice on increasing power and identifying important sequences holds here too.

We next prove consistency, again truncating the prior at $\tilde{L}$ and fixing $h$ and $\theta$.

**Summary of Proposition 22** *If $p_1^{(\tilde{L})} = p_2^{(\tilde{L})}$, then* $\text{BF} \to 0$ *as $N \to \infty$ in probability. Otherwise, if $p_1^{(\tilde{L})} \neq p_2^{(\tilde{L})}$, then* $\text{BF} \to \infty$ *in probability.* A proof is in Section G.2 and simulations in Section I.3.

## 6 Results

**Predicting sequences** We sought to evaluate BEAR models as compared to AR models on the task of predicting real nucleotide (nt) sequences. We considered eleven datasets of four different types: whole genome sequencing read data, single cell RNA sequencing read data (including from patient tumors), metagenomic sequencing read data (including from patient fecal samples) and full bacterial genomes from across the tree of life (Section K). Datasets ranged in total size from $\sim 10^7 - 10^{10}$ nt and in individual sequence length from $\sim 10^2 - 10^6$ nt (Table S1). 25% of data was randomly held out for testing, in the form of entire sequences (reads, genomes, etc., see Table S2); our goal was to evaluate BEAR models as density estimators, so we did not use masking (a common holdout strategy in natural language processing). We considered a linear AR model and a deep convolutional neural network (CNN) AR model with $> 10\times$ more parameters, both of which are common models used across a range of applications; we also designed a biologically-structured AR model which makes predictions based on a reference genome and a Jukes-Cantor mutation model

(Section L.1) [56, 67]. We then embedded each AR model to create a corresponding BEAR model. The BEAR models improve over the AR models in nucleotide prediction according to both perplexity (Table 1) and accuracy (Table S3) in all datasets, even when the model lag $L$ is held fixed for comparison (Section L.3).

In 10 out of 11 datasets, BEAR models increase nucleotide prediction accuracy from near chance values of $30 - 35\%$ (in the case of the linear and CNN models) to $78 - 95\%$, bringing genome-scale models into the realm of potential practical use (Table S3). The training time for BEAR models is essentially identical to that of AR models, aside from the time required to build the transition count matrix, which need only be done once before training all models (Fig. S13). Remarkably, the optimal lag $L$ chosen by empirical Bayes is often quite short, less than 20 nt (Table S4). The improvements offered by BEAR models that use an embedded AR model over the vanilla BEAR model are modest for datasets of this size; however, sequencing experiments are often designed to collect enough data for downstream analyses. We found in an example that, if sequencing coverage was $3\times$ instead of $100\times$, the improvement in prediction accuracy would have been greater than 10 percentage points instead of 0.1 (Section L.4; Fig. S14).

**Measuring misspecification** When conventional deep neural network methods fail to provide strong predictive performance, popular wisdom often ascribes the failure to too much model flexibility or not enough training data, especially in scientific applications. Examining the $h$ misspecification diagnostic in the BEAR models described above, we see that this is not the case here (Table 2). The large values of $h$ suggest that where the CNN fails it is not because of too much flexibility but rather too little: the model is not flexible enough to encompass the true data distribution, so it suffers from misspecification. Meanwhile, the reference-based model has only two learned parameters, but is less misspecified than the CNN in all but one dataset. This too runs counter to popular wisdom in machine learning, which often assumes that when principled, low-flexibility scientific models outperform deep neural networks it is thanks to their low variance in the small data regime.

Table 2: **Diagnostic $h$.** Abbreviations as in Table 1.

| Dataset | Lin. | CNN | Ref. |
|---|---|---|---|
| YSD1 | 5.528 | 5.461 | 4.183 |
| *A. th.* 1 | 2.765 | 2.756 | 2.990 |
| *A. th.* 2 | 2.643 | 2.633 | 2.326 |
| *A. th.* 3 | 3.969 | 3.964 | 1.598 |
| PBMC | 4.167 | 4.145 | 3.762 |
| HL | 4.050 | 4.038 | 3.581 |
| GBM | 4.172 | 4.154 | 3.238 |
| HC | 4.668 | 4.651 | - |
| CD | 3.096 | 3.094 | - |
| UC | 3.843 | 3.835 | - |
| Bact. | 0.010 | 0.003 | - |

**Generating samples** BEAR models are generative and can be used to sample new sequences. We sampled extrapolations from the end of a read sequence recorded in a plant (*A. thaliana*) whole genome sequencing experiment, and compared to an alternative non-probabilistic extrapolation method that is widely used in biology, local assembly (Fig. 3A; Section M). In this example the assembly algorithm SPAdes returns four possible assemblies, a relatively large number compared to other reads in the dataset (Fig. 3A stars) [5]. Samples from the BEAR model include these four possibilities, but also many more, some with higher probability. The distribution over possible nucleotide choices under the BEAR model is much wider than the number of assemblies would suggest: it has a perplexity of 1.4 per position (on average across samples) at the beginning of the extrapolation, and a perplexity of 2.7 at 50 nucleotides (Fig. 3B). These observations suggest that SPAdes, which does not provide a measurement of uncertainty, may not be capturing the full range of possible sequences.

**Visualizing data** Methods for learning local representations or features of biological sequences can be powerful tools for visualization and semisupervised learning [7]. One approach to extracting such representations is to learn a generative model $q(X_1, \ldots, X_{L+1})$ of kmers, for instance using a variational autoencoder. While such models are not autoregressive, the small size of the DNA alphabet makes it tractable to estimate the conditional $q(X_{L+1}|X_{1:L})$ by Bayes' rule, and this conditional can then be embedded into a BEAR model. We applied this strategy to probabilistic PCA. We visualized in low dimensions the inferred latent representation for a model trained on a single cell RNA sequencing dataset (HL), and were able to assign annotations to clusters, including those containing unmapped reads (Fig. 3CD; Section N). The BEAR model however raises the warning that the model is misspecified ($h = 4.836$), suggesting there may be richer latent structure yet to discover.

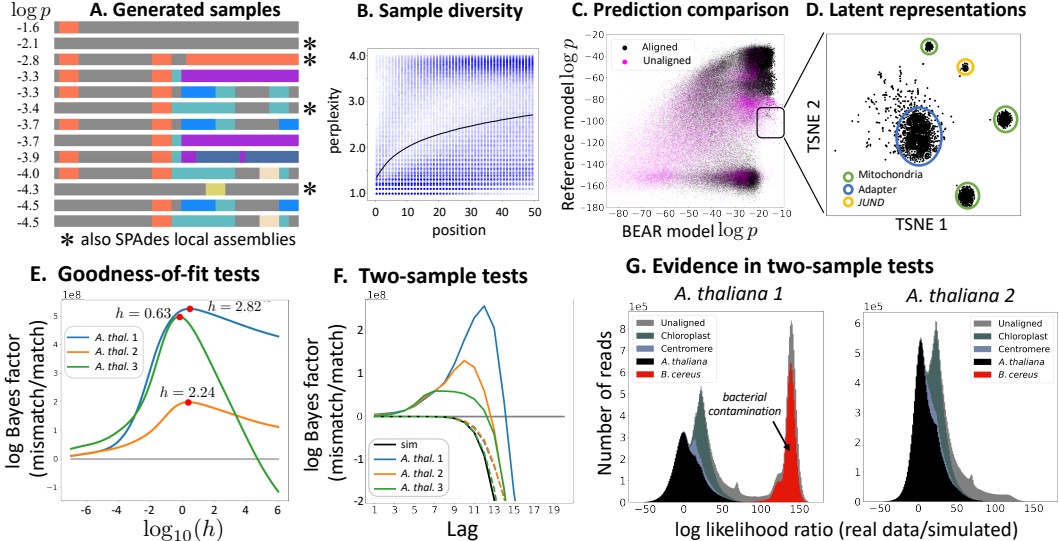

Figure 3: **Generation, visualization and testing.** (A) Sample extrapolations, colored to denote distinct paths through the $L$-mer de Bruijn graph. (B) Distribution of the perplexity of the next Markov transition under the BEAR model, for each position of the sampled extrapolations, with the per position average shown in black (Section M). (C) Log probability of each read in the HL dataset under the BEAR model and a model built from the reference transcriptome. Reads are colored by whether or not they map to the reference. (D) Latent representations of the reads highlighted in C, visualized using tSNE, with clusters annotated as likely coming from mitochondria, the sequencing adapter, or transcripts of the gene *JUND* (Section N). (E) Goodness-of-fit test Bayes factor as a function of hyperparameter $h$. (F) Two-sample test Bayes factor as a function of lag $L$. Black line compares simulated data to simulated data; dashed lines compare subsampled real data to subsampled real data; solid lines compare real data to simulated data. (G) Log probability of each read under the real data BEAR model minus the log probability under the simulated data BEAR model (Section O).

**Testing hypotheses** The question of when and how microbiomes change is widespread, but has in the past relied on summary statistics of sequencing datasets [44]. Schreiber et al. [64] studied changes in patient urine microbiomes before and after kidney transplant, and performed both unbiased metagenomic sequencing and diagnostic quantitative polymerase chain reaction (qPCR) for a specific virus associated with complications (JC polyomavirus). They found evidence of donor-to-recipient viral transmission in 5 cases out of 14. We applied the BEAR two-sample test to patients' metagenomic sequencing data before and after transplantation, using the vanilla Jeffreys prior and integrating over lags, in order to detect changes; the test rejects the null hypothesis in all 5 cases where there was transmission, and accepts the null hypothesis in all but one of the remaining 9 cases (Table S6; Section O.1). These results show, in a small example, that the two-sample test has sufficient power to detect microbiome changes in real data, and can be consistent with more specific tests.

We next applied BEAR hypothesis tests to evaluate generative models. We evaluated the reference-based AR model described above using the BEAR goodness-of-fit test. The test identifies considerable evidence (log Bayes factor $> 10^8$) for misspecification in each *A. thaliana* whole genome sequencing dataset, and this conclusion is robust to a wide range of $h$ values (Fig. 3E; Section O.2). Next, we evaluated a detailed simulation model (ART) that is intended to generate likely reads of a given reference genome [32]. The model lacks tractable likelihoods, so we use the BEAR two-sample test. When integrating over all lags, the test accepts the null hypothesis, suggesting that the simulation model is accurate; if we examine the test results for individual lags $L$ to increase power, however, we can see some evidence of differences (Fig. 3F; Section O.2). Note that as $L$ increases, there is a tradeoff: tests with larger lag can detect more subtle differences between the two distributions, but have less statistical power since they must consider a larger set of possible distributions. Thus the

Bayes factor first increases and then decreases with lag, reaching a peak at intermediate values where there is the most evidence of difference. To understand in detail the source of the detected differences between the data and the simulation model, we examined the conditional Bayes factor for individual reads, discovering clusters of reads that are poorly explained by the simulation model (Fig. 3G). One group mapped to chloroplasts, an organelle with its own genome that is variable in copy number; reads mapping to centromeres, an area of the plant genome for which the reference genome is considered unreliable, were also poorly explained by the simulation model. In one dataset we found a cluster of outliers that did not map to *A. thaliana* at all, and instead mapped to a common soil bacteria, *Bacillus cereus*, presumably a contaminant in the experiment (Fig. 3G, left). These results illustrate how BEAR hypothesis tests can be used not only for testing but also for detailed model criticism.

## 7   Discussion

In this article we proposed the nonparametric BEAR model, studied its theoretical properties, and developed algorithms and implementations for terabyte-scale inference. BEAR models substantially outperform standard AR models on a variety of datasets, and come with extensive theoretical guarantees, including for density estimation, misspecification detection, and hypothesis testing. BEAR models are closely connected to non-probabilistic genome analysis methods, such as de Bruijn graph assembly, but provide an alternative that is uncertainty-aware. Note, however, that BEAR models do not explicitly account for paired-end read information, or other sources of long-distance information; this is an important area for future work. While there has been little previous empirical or theoretical work in the machine learning literature on generative models of full genomic, transcriptomic or metagenomic sequences, we hope BEAR models provide a useful starting point.

## Acknowledgments and Disclosure of Funding

We thank Jean Disset for a small scale version of the kmer counting code, Rob Patro for crucial advice on large scale kmer counting, and Winnie Wang for illustrations used in the theory section of the supplementary material. We also thank Chris Sander, Elizabeth Wood, Tessa Green and all the members of the Marks Lab for discussion and ideas. E.N.W. is supported by the Fannie and John Hertz Foundation. D.S.M. is supported by the Chan Zuckerberg Initiative.

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
