# OpenReview forum: "A generative nonparametric Bayesian model for whole genomes"
_NeurIPS.cc/2021/Conference — NeurIPS 2021 Poster_

### Official Review · Reviewer_KMFd · 2021-07-14

**Rating:** 9
**Confidence:** 4

**Summary:**

In this study, authors have proposed a Bayesian embedded autoregressive model to tackle the problem of measuring and making DNA. They have implemented the model using a parametric auto-regressive model. The model is programmed in PYTHON language. They try to overcome challenges of  having enough flexibility to account for genomic complexity, interpretability to reach scientific conclusions, and scalability to train on billions of nucleotides. The authors have explored, theoretically and empirically, applications of BEAR models to a variety of statistical problems  including density estimation, robust parameter estimation, goodness-of-fit tests, and two-sample tests.

**Ethical Concerns:**

No concerns

**Main Review:**

The paper is well-written in English, and impressively organized. The major novelty of the model comes from nonparametrically perturb a parametric model, studying its theoretical properties, and developing algorithms and implementations for terabyte-scale inference. The model outperforms other standard models, however, it has some limitations such as not explicitly account for paired-
end read information, or other sources of long-distance information; But this model can be a starting point to address all these challenges. Authors have been honest about the limitations of their work.



**Time Spent Reviewing:**

6 hours

---

> ### Author Response · Authors · 2021-08-10
> **Response to Reviewer KMFd**
>
> We thank the reviewer for their thoughtful response.

---

### Official Review · Reviewer_mk3X · 2021-07-15

**Rating:** 6
**Confidence:** 3

**Summary:**

In this work, authors propose a new generative sequence model, called the Bayesian embedded autoregressive (BEAR) model. BEAR is a nonparametric Bayesian Markov model which embeds an autoregressive (AR) model into its prior. Theoretical guarantees are provided for statistical problems including density estimation, misspecification detection, and hypothesis testing, and experimental results indicate BEAR's superior performance on a variety of datasets.

**Limitations And Societal Impact:**

The authors have discussed the limitations in the discussion section, and the proposed method does not have any negative societal impacts.

**Main Review:**

\textbf{Strength}:
BEAR builds generative probabilistic models at the genome scale, which is able to detect and overcome misspecification problem. \\
In the paper, authors derive the asymptotic consistency guarantees and convergence rates for nonparametric density estimation, and prove consistency results for goodness-of-fit tests and two-sample tests, which provide theoretical guarantees of BEAR.\\
In the experimental evaluation, BEAR outperforms other autoregressive models on large-scale genome datasets.

\textbf{Weakness}:
I have several concerns regarding to the proposed method.
First, I wonder if BEAR can handle the long-range sequential data?\\
Next, I wonder if the parameters of BEAR are also interpretable? It would be great if the authors could discuss further about how to quantify the degree of misspecification introduced in the Section Robust parameter estimation, and how to select the optimal lag $L$. \\
Third, will the model be able to handle prior knowledge about some special elements on the genome sequence, such as transposable elements?


**Time Spent Reviewing:**

4

---

> ### Author Response · Authors · 2021-08-10
> **Response to Reviewer mk3X**
>
> We thank the reviewer for their thoughtful response.
>
> "I wonder if BEAR can handle the long-range sequential data?"
> BEAR can be trained on and generate arbitrarily long sequences; in the experiments, we apply BEAR to full bacterial genomes 6 million nucleotides long. In principle, BEAR can also describe arbitrarily long-range statistical dependencies, as proven in Section 3. In practice, the empirical Bayes inference procedure rarely selects lags larger than 20, suggesting that including longer range dependencies in the model leads to poorer generalization.
>
> "I wonder if the parameters of BEAR are also interpretable? It would be great if the authors could discuss further about how to quantify the degree of misspecification introduced in the Section Robust parameter estimation, and how to select the optimal lag L."
> The optimal lag is selected via an empirical Bayes procedure, taking advantage of the analytic marginal likelihoods of the BEAR model (Section 2). Misspecification is quantified also using an empirical Bayes procedure, optimizing the hyperparameter value h (along with theta) using minibatch estimates of the marginal likelihood (Section 2). The optimized value of h is the misspecification diagnostic (Section 4).
> Once the optimal value of h has been determined, we can interpret the BEAR model as in Section 4: if h is small, then the AR model is likely well-specified, and so the empirical Bayes estimate of theta is trustworthy. Further interpretation of the BEAR model parameters depends on the choice of autoregressive function f(theta). In the linear AR model, theta determines the linear coefficients on each position in each kmer. In the reference-based model described in Sections 6 and L.1, theta controls the probability of mutations from a reference sequence.
>
> "Third, will the model be able to handle prior knowledge about some special elements on the genome sequence, such as transposable elements?"
> There are a variety of ways of incorporating such prior knowledge into the BEAR model via careful design of the embedded AR model. For instance, we developed reference-based models that use a reference genome to inform predictions (Sections 6 and L). It is straightforward to build a model that incorporates reference transposable element sequences instead of (or in addition to) a reference genome, since the model can incorporate any sort of reference sequence information.

---

> ### Comment · Reviewer_mk3X · 2021-08-27
> **keep my original score**
>
> After reading author's response, I will keep my original score.

---

### Official Review · Reviewer_KhK3 · 2021-07-18

**Rating:** 7
**Confidence:** 3

**Summary:**

The paper introduces a generative Bayesian Markovian framework and a set of underlying models for DNA sequences to estimate the probability density of the current nucleotide given $L$ previous nucleotides $p(X_i|X_{i-L:i-1})$.
The authors investigate theoretically and empirically some applications of the framework such as density estimation of sequence data, parameter estimation and goodness-of-fit tests.

**Limitations And Societal Impact:**

The limitations of the BEAR model are briefly discussed in Section 7.
The social impact of genetic-related models is highlighted in Section A.

**Main Review:**

The paper is clearly written and well organized.

The novelty of the work lies in the theoretical description of the BEAR model and its applications to certain statistical problems for DNA modelling. The BEAR model is based on the Bayesian Markovian framework. It uses an embedded autoregressive (AR) model to estimate the hyperparamters of the Dirichlet prior distribution over the probability density of the kmers (nucleotide subsequences of length $L$).
The claims are adequately, theoretically and experimentally, supported.

#### Comments

1) I think the paper is too long for a conference venue. The submitted manuscript respects the 9 pages limit but each main section constantly refers the reader to other sections of the supplemental material. The presented models and the applications cannot be well understood without the support of this material. The theoretical results presented in the main text are a summary of a set of theorems and propositions about models, density estimation, parameter estimation and hypothesis testing that are detailed in 45 pages in the supplemental material. It is the same case for empirical results, which are supported by 17 pages in the supplemental material.

2) It is not clear whether the finite-lag Markov models described in Section D are new or does this name refer to an already known family of Markov models? From their definition, they are quite similar to variable-order Markov models. If not, what is the difference between both of these models? The variable-order Markov models have been used in nucleotide sequence analysis, e.g., in virus genome subtyping [1].

3) How does this work (embedding a parametric model in a Dirichlet prior) relate to the work of Tzu-Tsung Wong on Generalized Dirichlet distribution in a Bayesian analysis [2, 3]?

4) In the empirical evaluation, the authors designed “new” AR baseline models to embed them within the BEAR model and also to compare with. The results show that most of these baseline models are too weak and that their performance is lower or close to chance values. The comparison should also be done with other published methods or other deep architectures (RNNs, VAE) regardless of the AR framework (even RNNs could be considered as AR) [4, 5, 6]. In case the published methods cannot scale with terabyte-scale datasets (which I don't think is the case, e.g. metagenomics classification with linear methods [7] and UniRep for protein deep representation with mLSTM [8]), it will always be interesting to have comparison results on medium-size datasets.

#### Minor comments

1) Lines 23-25: Genome modelling using genetic variant matrix is usually used in GWAS studies. However, there is large repertoire of studies that models the whole genome at nucleotide resolution in different genomics analysis applications [4, 9]

2) Line 60: Rephrase: Crucial to our the theoretical and empirical analysis is the statistical setting.

#### References
* [1] Struck, D., et al. COMET: adaptive context-based modeling for ultrafast HIV-1 subtype identification. 2014.
* [2] Wong, T.T. Generalized Dirichlet distribution in Bayesian analysis. 1998.
* [3] Wong, T.T. and Liu, C.R. An efficient parameter estimation method for generalized Dirichlet priors in naïve Bayesian classifiers with multinomial models. 2016.
* [4] Im, Jinho, Byungkyu Park, and Kyungsook Han. A generative model for constructing nucleic acid sequences binding to a protein. 2019
* [5] Liang, Qiaoxing, et al. DeepMicrobes: taxonomic classification for metagenomics with deep learning. 2020.
* [6] Riesselman, Adam J., John B. Ingraham, and Debora S. Marks. Deep generative models of genetic variation capture the effects of mutations. 2018.
* [7] Vervier, K., et al. Large-scale machine learning for metagenomics sequence classification. 2016.
* [8] Alley, Ethan C., et al. "Unified rational protein engineering with sequence-based deep representation learning. 2019.
* [9] Lim, D., and Blanchette M. EvoLSTM: context-dependent models of sequence evolution using a sequence-to-sequence LSTM. 2020.


**Time Spent Reviewing:**

12

---

> ### Author Response · Authors · 2021-08-10
> **Response to Reviewer KhK3**
>
> We thank the reviewer for their thoughtful response.
>
> "I think the paper is too long for a conference venue..."
> We sympathize with the reviewer's concern, however, we believe that the NeurIPS audience is the appropriate audience for the paper, as it is a blend of theoretical and applied machine learning, statistics, and biology.
>
> "It is not clear whether the finite-lag Markov models described in Section D are new or does this name refer to an already known family of Markov models? From their definition, they are quite similar to variable-order Markov models. If not, what is the difference between both of these models? ..."
> In this paper, we refer to a Markov model's "order" as its "lag"; the set of finite-lag Markov models described as \mathcal{M} in Section D is exactly the set of variable-order Markov models.
>
> "How does this work (embedding a parametric model in a Dirichlet prior) relate to the work of Tzu-Tsung Wong on Generalized Dirichlet distribution in a Bayesian analysis [2, 3]?"
> We use a standard Dirichlet prior, with its concentration parameters controlled by a parametric model, rather than a Generalized Dirichlet distribution.
>
> "In the empirical evaluation, the authors designed “new” AR baseline models to embed them within the BEAR model and also to compare with… The comparison should also be done with other published methods or other deep architectures (RNNs, VAE) regardless of the AR framework (even RNNs could be considered as AR) [4, 5, 6]..."
> We do not mean to suggest that the AR baseline models we considered are new: the linear and CNN models are standard model architectures, and the reference-based model is a natural extension of existing models. We also studied a VAE model, with one layer and nonlinearity (the probabilistic PCA model in the Visualizing Data section and Section N); we found that the conditional distribution of the model has a perplexity of 4.27 on the Hodgkin's lymphoma dataset, and a perplexity of 1.39 when embedded in the BEAR. It is thus worse than the other AR baseline models, including the linear AR model. We will clarify these points further in the text.
> Other large scale sequence models are not comparable on the genome-scale read datasets considered in the paper. The metagenomics classification method in [7] is highly scalable, but it is a classifier and not a generative sequence model, and thus incomparable. The Unirep model is a protein model rather than a nucleotide model, and requires orders of magnitude more resources: (1) Unirep took roughly 3 weeks to train on a dataset roughly the same size (~8 billion amino acids) as the single cell sequence datasets we studied (20-30 billion nucleotides), while BEAR models train in less than 12 hours using less hardware (two Tesla K80 GPUs rather than 4), (2) our scalable implementation of the BEAR model relies crucially on recent advances in large-scale DNA kmer counting, in particular KMC3 (Kokot et al. 2017), and comparable methods are not available for proteins. Scalability is a particular concern in our suggested applications such as two-sample testing, where efficient evaluation of multiple samples is important.
>
> "... there is large repertoire of studies that models the whole genome at nucleotide resolution in different genomics analysis applications [4, 9]"
> We have not found many examples of generative, nucleotide resolution whole genome models. For instance, reference [4] describes a generative model trained on sequences of length 13-40 that are (primarily) putative transcription factor binding sites. By contrast, BEAR models can be trained on complete genomes millions or billions of nucleotides long. Reference [9] describes a model of point mutations based on whole-genome alignments, but similar to standard mutation models such as Jukes-Cantor, it cannot generate the more complex structural variation seen in many real datasets.
>
> "Rephrase: Crucial to our the theoretical and empirical analysis is the statistical setting."
> We will fix this typo.

---

> > ### Comment · Reviewer_KhK3 · 2021-09-07
> > **Paper too long with many details + references required**
> >
> > Thank you for the detailed answer.
> >
> > 1- I am not completely satisfied with the answer of the first concern. I looked at the paper another time. It's really long with a lot of important details in its supplementary material (SM). Even the main models (AR and BEAR) are defined in the main manuscript, there are several variations of these models defined in the SM depending on the task (prediction, hypothesis testing, etc. ). Thus, it is difficult to follow the details of a specific model with its applied task and its evaluation.
> >
> > 4- If the linear and CNN autoregressive models are not considered here new for generative models using nucleotide data, please indicate clearly the reference papers where they were specified and used in the same context/tasks as your models.

---

> > > ### Author Response · Authors · 2021-09-13
> > > **Reviewer Reply**
> > >
> > > We thank the reviewer for their reply. In response to concern 4, we will add additional references to the most closely related uses of these AR models, in particular the compression literature discussed above with Reviewer MWYx.

---

### Official Review · Reviewer_MWYx · 2021-07-19

**Rating:** 7
**Confidence:** 3

**Summary:**

The authors propose a generative sequence model, Bayesian embedded autoregressive (BEAR), which is non-parametric as opposed to parametric autoregressive models to address the misspecification problem in neural networks. The BEAR model is a Bayesian Markov model with an embedded autoregressive model into its prior. Using parametric model output as a prior allows BEAR to be used for non-parametric density estimation which makes asymptotic consistency possible. They also show that the learned concentration hyperparameter for a prior dirichlet distribution represents the amount of misspecification in the embedded autoregressive model (or other neural network that was embedded) that can be used as a warning when the model can not be trusted. Then, they prove that absolute performance of the distribution predicted by some model can be measured through testing whether it matches the optimal BEAR model.

They theoretically prove that the BEAR model is more expressive than the naive AR model and that it converges to the true distribution as the size of the dataset increases. BEAR models are scalable and uncertainty-aware, which make them a useful generative model for full biological sequences.

**Limitations And Societal Impact:**

Yes

**Main Review:**

EDIT 2021-08-19: Changed score to 7.

The paper addresses the interesting problem of sequence generation with a convergence guarantee, and is scalable to whole genome sequences which make it practically used as a generative model. It is well-written and all claims are followed by theoretical proofs and simulation results. My main concern is that it does not reference or compare to the related literature on sequence compression.

Major comments:

- What is the utility of a generative model of biological sequences? The authors say that it would be useful to have a generative model, but not what it might be used for. The closest experiment to being practical is in section "Testing hypotheses", in which the authors design a statistical test for whether two sets of sequences are generated from the same distribution. Is this the main aim of BEAR?
- Density estimation is closely related to compression. There is a large literature of nucleotide sequence compression methods, such as the following, which the authors should cite and compare and contrast.
	- https://academic.oup.com/gigascience/article/9/11/giaa119/5974977
	- https://academic.oup.com/bib/article/16/1/1/240815
- The manuscript includes a huge 62-page supplement. Much of the paper is unclear without reference to the supplement, including many of the experiments. The manuscript would be improved with a narrower focus, particularly on those tasks that are practically relevant.
- In particular, the formulation of fk(θ) is not in the main manuscript, although it is very needed to know the parameters that are going to be learned during training. For example, what is B when β* > 0 in AR model if it is not learned during fitting? What is its value during sample generation?
- In the “measuring misspecification” subsection in the results section, the relation between the misspecification and flexibility is not clear. The misspecification value has been shown to be proportional to a divergence between AR model and true distribution, however it is not clear how the divergence is related to the flexibility of the model.
- In the two-sample test experiment (evaluating a simulation model, ART), it has been shown that differences between real and simulation data can be observed when the test is done for individual lags. The behavior of bayes factor in terms of lag is not clear, for example why it increases and then decreases when comparing simulation and real data.

**Time Spent Reviewing:**

6

---

> ### Author Response · Authors · 2021-08-10
> **Response to Reviewer MWYx**
>
> We thank the reviewer for their thoughtful response.
>
> "My main concern is that it does not reference or compare to the related literature on sequence compression… Density estimation is closely related to compression."
> The reviewer's connection to the compression literature is insightful, and we will add discussion of this connection in the related work section. The compression literature contains a number of example applications of AR and Markov models for density estimation (e.g. https://ieeexplore.ieee.org/document/4148743, https://academic.oup.com/gigascience/article/9/11/giaa119/5974977, https://journals.plos.org/plosone/article?id=10.1371/journal.pone.0021588). These methods, like other AR models, can be embedded into BEAR models for improved statistical performance.
>
> "What is the utility of a generative model of biological sequences?..."
> Generative models of biological sequences have wide and growing utility. One important use case is for sequence design: sequences sampled from the model can be constructed in the lab through e.g. synthesis or genome editing. Design methods based on generative models have been proposed for antibodies (https://www.nature.com/articles/s41467-021-22732-w), enzymes (https://science.sciencemag.org/content/369/6502/440), regulatory regions (https://icml-compbio.github.io/icml-website-2020/2020/papers/WCBICML2020_paper_46.pdf) and other important classes of protein and DNA sequences, and the practical success of these design strategies depends in large part on the quality of the model. Another important (related) use case is for fitness estimation, as state-of-the-art fitness estimators are based on generative probabilistic models; such models have been applied to predict the effects of mutations on protein function and human disease (https://www.nature.com/articles/s41592-018-0138-4, https://www.biorxiv.org/content/10.1101/2020.12.21.423785v1). The BEAR model is both a powerful generative probabilistic model itself (as demonstrated in the density estimation section), and a powerful tool for rigorously evaluating other generative models (as demonstrated in the robust parameter estimation and hypothesis testing sections). We will add further discussion of the uses of generative sequence models in the introduction.
>
> "The manuscript includes a huge 62-page supplement... The manuscript would be improved with a narrower focus, particularly on those tasks that are practically relevant."
> We believe each of the statistical tasks considered is practically relevant, as discussed above. While we sympathize with the concern about length, we see each problem and its underlying theoretical justification as closely related.
> "In particular, the formulation of fk(θ) is not in the main manuscript, although it is very needed to know the parameters that are going to be learned during training. For example, what is B when β* > 0 in AR model if it is not learned during fitting? What is its value during sample generation?"
> We explore a wide variety of parametric functions f(theta), which are detailed in the supplementary material due to manuscript length restrictions: Section I describes the functions f(theta) used in the simulation experiments, and Section L describes the functions used in the experiments with real data. In the experiments shown in Figure 2, we sample data from an AR model with parameters A and B specified in Section I.1.1. We then fit a BEAR model with B = 0 because we want to evaluate the behavior of the BEAR model when the embedded parametric model f(theta) is misspecified. This demonstrates a crucial advantage of the BEAR model, namely that it is a consistent nonparametric density estimator and thus not susceptible to misspecification. We will clarify this point further in the text.
>
> "In the “measuring misspecification” subsection in the results section, the relation between the misspecification and flexibility is not clear..."
> Flexibility describes the size of the set of distributions that can be fit by the model by varying its parameters. Misspecification refers to the case where this set does not include the true data distribution, so it is natural to address misspecification by increasing flexibility. We will add clarification of this point.
>
> "In the two-sample test experiment ... The behavior of bayes factor in terms of lag is not clear, for example why it increases and then decreases when comparing simulation and real data."
> As the lag increases, there is a tradeoff: tests with larger lag can detect more subtle differences between the two samples, but have less statistical power since they must consider a larger set of possible distributions. This tradeoff results in the Bayes Factor reaching its maximum at an intermediate lag value. We will add a sentence clarifying this point.

---

> > ### Comment · Reviewer_MWYx · 2021-08-19
> > **2021-08-19**
> >
> > Thank you to the authors for the response.
> >
> > Sequence compression: Including a review of sequence compression methods is an improvement. However, the authors evaluate BEAR with respect only to simple AR model. What about a more realistic AR or Markov model used by sequence compression methods?
> >
> > My concerns regarding the length of the supplement remain. My other comments have been addressed. I changed my score to "7: Accept".

---

> > > ### Author Response · Authors · 2021-08-23
> > > **Reply to Reviewer MWYx**
> > >
> > > We thank the reviewer for their reply. While we did evaluate deep neural AR architectures, we agree with the reviewer that specific comparisons to the precise architectures used in the compression literature could be instructive going forward.

---

### Decision · Program_Chairs · 2021-09-27

**Decision:**

Accept (Poster)

**Comment:**

All four reviewers recommend accepting the submission, but three reviewers are not highly confident in their assessments and I am not highly confident in the other reviewer’s assessment; there is a disconnect between what this reviewer wrote and what rating this reviewer assigned. Therefore, I reviewed this submission carefully myself. The submission is well written, novel, and makes an important contribution to Bayesian modeling for genomic sequence data; I recommend accepting it.